# Possible Locking Shock Time in 2–48 Hours

**Tao Chen** [1,*,†], **Lei Li** [1,2,†], **Xiaoxin Zhang** [3], **Chi Wang** [1], **Xiaobing Jin** [4], **Han Wu** [5], **Shuo Ti** [1], **Shihan Wang** [1,2], **Jiajun Song** [6], **Wen Li** [1], **Jing Luo** [1], **Chunlin Cai** [1], **Xuemin Zhang** [7], **Shi Che** [7], **Xiaodong Peng** [1] and **Xiong Hu** [1]

1  State Key Laboratory of Space Weather, National Space Science Center, Chinese Academy of Sciences, Beijing 100190, China
2  School of Earth and Planetary, University of Chinese Academy of Sciences, Beijing 100049, China
3  National Center for Space Weather, China Meteorological Administration, Beijing 100081, China
4  Sichuan Meteorological Disaster Defense Center, Chengdu 610072, China
5  College of Science, Southern University of Science and Technology, Shenzhen 518055, China
6  Institute of Electrical Engineering, Chinese Academy of Sciences, Beijing 100190, China
7  Institute of Earthquake Forecasting, China Earthquake Administration, Beijing 100036, China
*  Correspondence: tchen@nssc.ac.cn; Tel.: +86-13910421558
†  These authors contributed equally to this work and should both be considered first authors.

**Abstract:** An hourly scale precursor of inland earthquakes (EQs) is revealed in this paper. Several EQ cases in China have been reported. As indicated by a table listing 23 inland EQs and their shock time, epicenter location, magnitude, near-epicenter weather conditions, precursor start time and precursor duration, when the weather conditions are fair near the epicenter, an anomalously negative atmospheric electrostatic signal is readily observable approximately 2–48 h before the EQ occurs. Moreover, a successful single-station alarm for nearby moderate-magnitude EQs is demonstrated, and a possible mechanism for the precursor signal is proposed. The change in the electrostatic field during an EQ process is explained as the release of radioactive gases from the subsurface into the atmosphere via large (regional-scale) preexisting microfractures in the rock at the source depth. These gases considerably ionize the atmosphere, and the separated positive and negative ions establish a special macroscopic electric field. The final critical stage of 2–48 h before an EQ may indicate a stable tectonic process.

**Keywords:** inland earthquakes; reliable precursor; atmospheric electric field; hourly scale features; shock time

## 1. Introduction

Due to the devastating consequences of strong earthquakes (EQs), the identification of reliable precursors is of paramount importance. There are many kinds of parameter anomalies that could be considered EQ precursors [1–3]: geostress changes; underground water level and other fluid changes; the geoelectric field, geoconductivity and geomagnetic disturbance records; gravitational anomalies; surface deformations; very large gas fluxes out of the crust; variations in radon gas; temperature variations at the Earth's surface; air temperature variations; variations in the relative humidity in the air; anomalous fluxes of the latent heat of evaporation; extraordinary vertical profiles of air temperature and humidity; linear cloud anomalies; anomalies of radio wave propagation in very low frequency (VLF), high frequency (HF) and very high frequency (VHF) bands; extraordinary concentrations and distributions of aerosols; anomalies of the outgoing longwave radiation (OLR) energy flux; local (in situ) anomalies of space plasma parameters (concentrations of ions and electrons, ion and electron temperatures, mass compositions and concentrations of the major ions); extraordinary extremely low frequency (ELF) and VLF emissions measured onboard satellites; quasiconstant magnetic and electric fields; extraordinary particle precipitation fluxes for different energy bands; vertical profiles of the electron concentration; and extraordinary total electron content (TEC) changes detected by GPS data processing [1].

In addition, atmospheric electrostatic field monitoring has recently become more popular. Atmospheric electric field anomalies prior to EQs have been widely studied [4–11]. Since the Tangshan EQ in 1976, China has set up several monitoring stations of atmospheric electric fields, and some obvious anomalous cases have been observed [5,10]. Under normal conditions, the atmospheric electric field is mainly modulated by meteorological conditions and solar activities. On a fair day without the influence of human activities, if a negative anomaly of the atmospheric electric field appears, it is likely to be followed by an EQ event.

There are various methods for measuring the atmospheric electric field; commonly used methods involve ground-based rotating electric field meters, roller electric field meters, sphere-borne double-ball electric field meters and microrocket electric field meters. In the past, the most common methods were potential probes and burning fuses, and these methods are still used by some observatories today. The most widely used method today is the electric field mill (EFM), which provides good exposure to atmospheric electric fields. It usually consists of one or more electrodes that are alternately shielded and exposed to the atmospheric electric field. Long-term observational studies of atmospheric electric fields have been executed since the 1980s by different research teams [12–14] using EFM100 in Sichuan, Yunnan, Beijing, Hebei, etc. The atmospheric electric field meters that were used for measurements in this paper are also EFM100 with a measurement accuracy <5% and resolution of 10 V/m.

Recent observations [12,15] revealing the hourly scale characteristics of inland EQ precursors indicated that detecting these precursors is feasible. Compared with preseismic anomalies in low-frequency geoacoustic emissions and in ultralow frequency (ULF) and VLF electromagnetic emissions, preseismic anomalies within the atmospheric electrostatic field occur closer in time to strong inland EQs. Omori et al. [11] noted that anomalous radon emissions trigger great decreases in the atmospheric field of the lower atmosphere (from the ground to an altitude of 2 km), as observed around the time of the Kobe EQ in 1995. They further suggested that the behavior of radon in terms of the atmospheric electrostatic process can explain the seismic precursors observed near the ground. Omori et al. [11] further suggested that the behavior of radon in terms of the atmospheric electrical quasistatic process can explain seismic precursors observed near the ground. Choudhury et al. [4] described the characteristics of the vertical atmospheric electrostatic field as negative 7–12 h before an EQ according to the statistics of 30 EQ events of various classes over northern India. Smirnov [9] reported more than one hundred cases featuring negative vertical Ez anomalies approximately one day in advance for Ms 4–6 EQs, but similar to the relationship between Ez and magnitude, the Ez value and epicentral location do not exhibit an obvious relationship. Furthermore, previous studies indicated that the near-surface atmospheric electrostatic field is unique in that it becomes anomalously negative (the downward direction of Ez is defined as positive under fair-weather conditions) just before some EQs under fair-weather conditions [10,16], but these studies did not summarize the features of the electrostatic field at an hourly scale. Thus, this paper examines the hourly scale features of anomalously negative signals of the atmospheric electrostatic field as precursors. These signals could be the most stable and reliable indicator warning of an imminent EQ several hours to one day in advance and could therefore be leveraged to alert people of an impending emergency.

## 2. Temporal Characteristics of Near-Surface Atmospheric Electrostatic Abnormalities before an EQ

First, we describe four abnormal atmospheric electrostatic signal cases that always appear several hours to one day before an inland EQ occurs and present a list of inland EQs with their shock time, epicentral location, magnitude, weather conditions, precursor beginning time and precursor duration.

Figure 1 shows the four negative atmospheric electrostatic field Ez cases before four EQs (the Beijing Ms 3.0, Rongcounty Ms 4.7, Changning Ms 6.0 and Wenchuan Ms 8.0 EQs). The red circles indicate the epicenters and the red stars represent the stations. One case

is shown in Figure 1a. Before the Beijing EQ, which occurred on 14 April 2019, in Beijing, China, the weather was identified as fair. An abnormal phenomenon was observed when the normal positive Ez became negative (approximately −100 V/m) at 09:00 in the morning LT (3.8 h before the EQ), as shown in Figure 1a. A medium-magnitude EQ was predicted to occur several hours later based on previous analytic results. Later, an Ms 3.0 EQ was confirmed to have occurred at 12:47 LT with the epicenter 40 km north of where the scientific instrument was located.

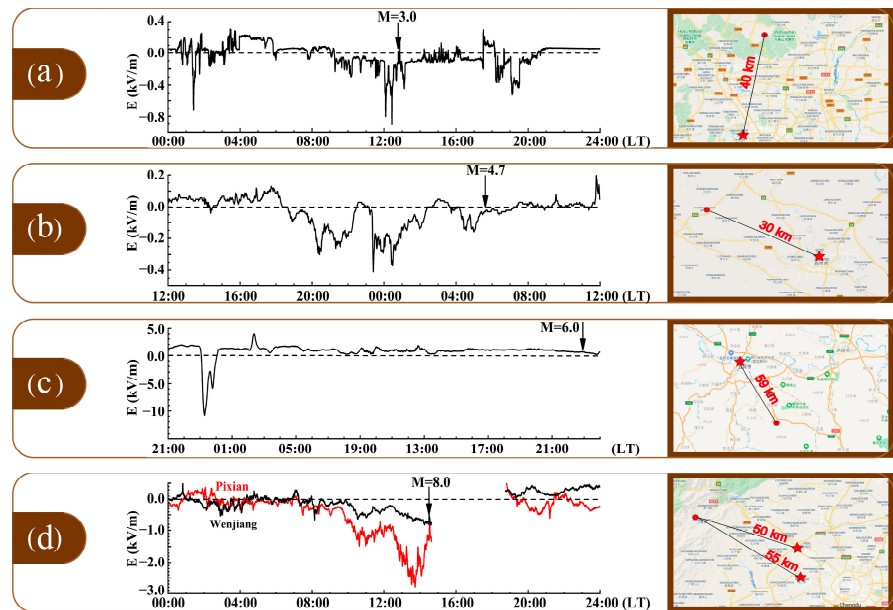

**Figure 1.** Negative atmospheric electrostatic field Ez before earthquakes. (**a**) Beijing Ms 3.0 earthquake on 14 April 2019, with a station located 40 km from the epicenter. (**b**) Rongxian Ms 4.7 earthquake on 24 February 2019, with a station located 30 km from the epicenter. (**c**) Wenchuan Ms 8.0 earthquake on 12 May 2008, with two stations located 50 and 55 km from the epicenter. (**d**) Changning Ms 6.0 earthquake on 17 June 2019, with a station located 50 km from the epicenter. The data gap in panel (**c**) is due to a power outage. In the maps on the right, the stars denote the stations and the circles denote the epicenters.

Similarly, as shown in Figure 1b, the Rongxian Ms 4.7 EQ occurred at 05:38 LT on 24 February 2019, in Sichuan Province (longitude = 104.49° E, latitude = 29.47° N), and Ez became negative 11 h earlier under fair air conditions at Zigong Station, which was situated 30 km from the epicenter. Ez suddenly decreased to a minimum of −410 V/m at 23:24 LT (the maximum negative Ez value). Although there were two short intervals (tens of minutes) of positive Ez values, Ez showed a negative anomaly for more than ten hours before the EQ began.

Figure 1c also shows that the Changning Ms 6.0 EQ occurred at 23:30 LT on 17 June 2019, in Sichuan Province (longitude = 104.49° E, latitude = 29.47° N) and the Ez became negative approximately 23.5 hours before the EQ under fair weather conditions at the Yibin Station, which was situated 50 km from the epicenter. Ez suddenly decreased to a minimum of −10,000 V/m at 23:30 LT (the maximum negative Ez value). The negative Ez signal lasted approximately 70 min, and then Ez recovered to normal positive values. Twenty-three hours later, EQ occurred. It is important to note that there were also significant positive Ez signals before that EQ, but these signals were not considered because they are not ubiquitous; i.e., not every EQ involves a positive Ez signal several hours beforehand. Hence, positive Ez signals may be irrelevant.

Another case shown in Figure 1d is that for the Wenchuan Ms 8.0 EQ that occurred at 14:28 LT on 12 May 2008 (longitude = 103.32° E, latitude = 31.00° N). Despite a data gap, the Ez signals in Pixian and Wenjiang Counties showed negative values for 7 h before

the EQ. Note that the weather in Wenchuan, Pixian and Wenjiang counties was very fair all day as well. The peak negative values of Ez at the Pixian and Wenjiang stations were −2750 V/m and −750 V/m, respectively. The Pixian Station is 50 km from the epicenter, while the Wenjiang Station is 55 km from the epicenter.

There are many other examples, such as these four cases, where abnormal atmospheric electrostatic signals appear several hours to one day before an inland EQ occurs. A list of 23 inland EQs with their shock times, epicentral locations, magnitudes, weather conditions, precursor starting times and precursor durations is presented in Table 1.

**Table 1.** Inland EQs reported in the literature.

| Date | Epicenter | Magnitude | Depth | Advance Time | VEF Depth | VEF Duration | Weather | Citation |
|------|-----------|-----------|-------|--------------|-----------|--------------|---------|----------|
| (yyyy/mm/dd) | | (M) | (km) | (h) | (−kV/m) | (h) | | |
| 1976/08/23 | Sichuan–Gansu border region, China | 7.2 | 23 | 23 | 17 | 26.5 | Fair | Hao et al. (2000) [5] |
| 1986/8/30 | Romania | 7.2 | 132.3 | 18.5 | 0.28 | 2 | Fair | Nikiforova and Michnowski (1995) [17] |
| 1992/3/5 | Off the east coast of the Kamchatka Peninsula, Russia | 6.4 | 45.2 | 9.5 | 0.4 | 1 | Fair | Rulenko et al. (1992) [18] |
| 1997/9/24 | Near the east coast of the Kamchatka Peninsula, Russia | 4.4 | 33 | 19 | 0.8 | 1.7 | Fair | Smirnov (2019) [19] |
| 1999/9/6 | Off the east coast of the Kamchatka Peninsula, Russia | 5 | 55.4 | 26 | 0.3 | 8 | Fair | Smirnov et al. (2017) [20] |
| 1999/9/9 | Kuril Islands | 5.5 | 33 | 27 | 0.3 | 7 | Fair | Smirnov et al. (2017) [20] |
| 1999/9/18 | Off the east coast of the Kamchatka Peninsula, Russia | 6 | 60 | 29.5 | 0.7 | 5 | Fair | Mikhailova and Mikhailov (2004) [21]; Mikhailova et al. (2013) [22]; Smirnov et al. (2017) [20] |
| 1999/10/24 | Off the east coast of the Kamchatka Peninsula, Russia | 5.3 | 44.4 | 2.5 | 0.4 | 1.1 | Fair | Smirnov (2019) [19] |
| 2002/10/16 | Off the east coast of the Kamchatka Peninsula, Russia | 6.2 | 102.4 | 34 | 0.3 | 3 | Fair | Mikhailov et al. (2006) [23] |
| 2008/5/12 | Eastern Sichuan, China | 7.9 | 19 | 7 | 2.75 | 7 | Fair | Chen et al. (2021b) [15] |
| 2009/7/26 | Andaman Islands, India region | 5.2 | 10 | 8.3 | 0.714 | 1 | Fair | Choudhury et al. (2013) [4] |
| 2009/12/12 | Maharashtra, India | 5 | 10 | 5.3 | 0.548 | 0.9 | Fair | Choudhury et al. (2013) [4] |
| 2009/12/13 | India | 5.1 | 10 | 11.3 | 0.633 | 0.9 | Fair | Choudhury et al. (2013) [4] |
| 2010/5/1 | Andaman Islands, India region | 4.6 | 12 | 10.1 | 0.834 | 0.9 | Fair | Choudhury et al. (2013) [4] |
| 2010/9/10 | Meghalaya, India region | 4.8 | 15 | 11.3 | 0.804 | 1.2 | Fair | Choudhury et al. (2013) [4] |
| 2010/12/12 | India | 4.8 | 15 | 6.1 | 0.457 | 1.1 | Fair | Choudhury et al. (2013) [4] |
| 2011/2/12 | India | 4 | 10 | 7.7 | 0.515 | 0.9 | Fair | Choudhury et al. (2013) [4] |
| 2012/4/24 | Nicobar Islands, India region | 5.5 | 10 | 12.3 | 0.684 | 1 | Fair | Choudhury et al. (2013) [4] |

**Table 1.** *Cont.*

| Date | Epicenter | Magnitude | Depth | Advance Time | VEF Depth | VEF Duration | Weather | Citation |
|---|---|---|---|---|---|---|---|---|
| (yyyy/mm/dd) | | (M) | (km) | (h) | (−kV/m) | (h) | | |
| 2012/5/11 | India | 5.4 | 20 | 12.7 | 0.562 | 1.3 | Fair | Choudhury et al. (2013) [4] |
| 2018/12/16 | 24 km WSW of Zhongcheng, China | 5.4 | 27.34 | 24 | 0.385 | - | Fair | Smirnov et al. (2017) [20] |
| 2019/6/17 | 6 km WNW of Yanling, China | 6 | 16 | 23 | 11 | 1.1 | Fair | Chen et al. (2021b) [15] |
| 2010/6/13 | Nicobar Islands, India region | 7.8 | 10 | 14.1 | 1.385 | 0.7 | Fair | Choudhury et al. (2013) [4] |
| 2021/4/16 | Luanzhou, Hebei, China | 4.3 | 9 | 6.6 | 2.7 | 8.6 | Fair | Chen et al. (2022) [24] |

Table 1 shows the details of the inland EQs that are recorded in the literature, including the three key elements of the EQs (shock time, epicentral location and magnitude) and characteristics of the electrostatic field. Among these EQ cases, the magnitude was Ms 3–8 and the proportions of sensed EQs ($4.5 >$ Ms $\geq 3.0$), medium-strong EQs ($6.0 >$ Ms $\geq 4.5$) and strong EQs (Ms $\geq 6.0$) were 13%, 52% and 35%, respectively. All these inland EQs occurred on fair-weather days with negative electrostatic field anomalies appearing 2–48 h before the EQ. Most of these anomalously negative Ez signals lasted for a short time (approximately 1 h). The values of the negative electrostatic field signals ranged from a few hundred to tens of thousands of volts per meter, which may have been related to the distance between the station and epicenter or the local geological structure. All 23 cases had a shock time of 2–48 h, and 83% of the anomalies occurred no more than one day before the EQ, which is a critical and effective time scale for EQ disaster prevention.

According to these case studies and EQ list analysis, it is considered possible that if an inland EQ is impending, the crustal movements near the related faults cause the near-surface atmospheric electrostatic field to become abnormal. Additionally, other authors [4,9] have found that precursors appear at a time scale of 2–48 h before an EQ. With these observation-based descriptions from all over the world, it is likely that fair-weather anomalously negative Ez signals ubiquitously appear 2–48 h before an inland EQ occurs.

## 3. A Successful Single-Station Alarm for Nearby Moderate-Magnitude Earthquakes

To examine and explain the relationship between the electric field signal and crustal activity, Chinese Meridian Project researchers have established a regional monitoring network of the atmospheric electric field in Sichuan Province, China. All these stations use EFM100 electric field meters with a measurement range of $\pm 50$ kV/m, a measurement accuracy of <5% and a resolution of 10 V/m. Mianning Station (longitude = 102.17° E, latitude = 28.55° N), where a period of negative anomalies appeared before the Mianning EQ, is part of this monitoring network.

The negative atmospheric electrostatic field Ez signal before the Ms 3.6 Mianning EQ, which occurred at 20:44 LT on 4 October 2022 (longitude = 102.14° E, latitude = 28.91° N), is shown in Figure 2. The red circles and stars represent the same features as in Figure 1. The distance between the station and the epicenter is 40 km. The electric field started to show a significant negative anomaly at 20:10 LT on October 2nd (47.4 h before the EQ), and the negative anomaly lasted for approximately 7 h. The negative anomaly peak was -4.42 kV/m at 22:37 LT on October 2nd. Based on the meteorological data obtained from the Central Meteorological Observatory, the weather was very fair during the period of the negative electrical field.

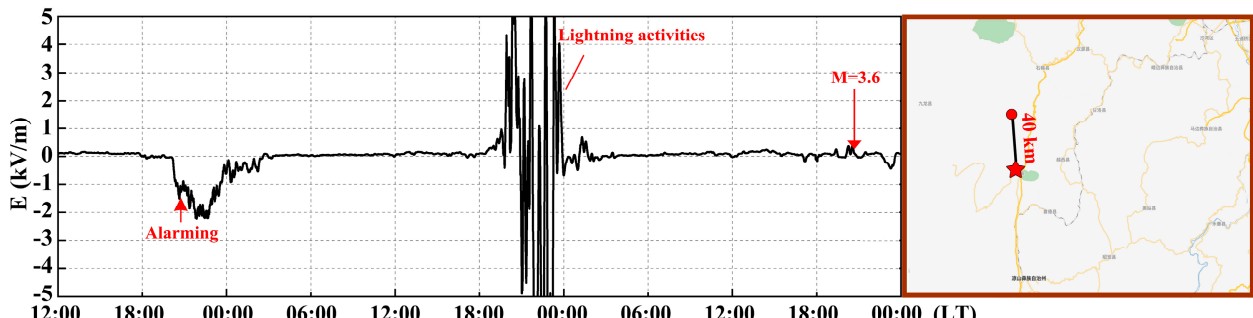

**Figure 2.** Negative atmospheric electrostatic field Ez signal before the Mianning M3.6 earthquake. The circle indicates the epicenter and the star indicates the Mianning Station.

There is a very obvious period of electric field signal anomalies caused by lightning activities before the EQ shown in Figure 2, with a sharp jump in positive and negative values. Compared with the previous meteorological conditions, there was 0.4–8.5 mm of continuous precipitation in the period from 21:00 LT on October 3rd to 04:00 LT on October 4th, with a maximum relative humidity of 95% and 10 min average visibility not exceeding 10 km. This confirmed that the electric field signal resulted from the effect of lightning activities.

To provide an early warning of an EQ by detecting abnormal signals within the atmospheric electric field during fair weather, researchers have used artificial intelligence recognition technology to build a system that can compare meteorological conditions and electric field values to discern whether there are negative anomalies. The system currently uses a simple algorithm that still needs to be improved. At 22:33 LT on October 2nd (46.2 h before the EQ), this system sent an alarm, which indicated the presence of negative anomalies in the fair weather and warned people to stay alert. This was an example of a successful single-station alarm indicating a nearby moderate-magnitude EQ.

## 4. Discussion

Based on five recent electrical field negative anomaly cases before EQs in China and the 23 cases in Table 1, it can be concluded that the atmospheric electric field always shows negative anomalies 2–48 h before significant inland EQs (Ms ≥ 3.0) and a shock time of 2–48 h may be largely possible. Based on our current statistical results, the time interval between the occurrence of most EQs and atmospheric electric field anomaly signal does not exceed 24 h; it means in most cases that people have no more than 24 h to leave their work and homes around the epicenter. A possible physical mechanism for why the electric field negative anomaly signal occurs so close to the EQ occurrence is depicted in Figure 3.

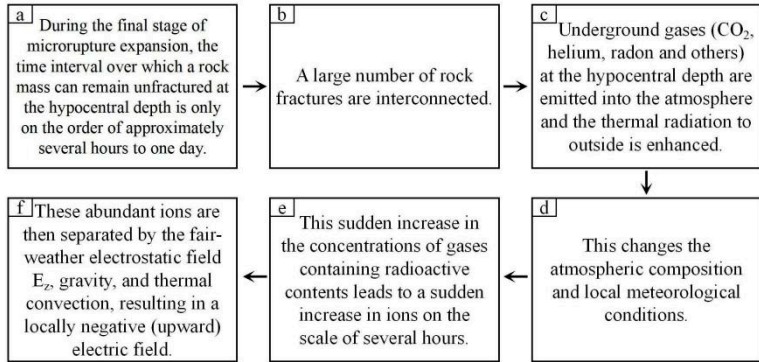

**Figure 3.** Logical flow diagram illustrating the hourly scale physical processes leading up to an earthquake. Subfigures a-f are the sequential physical logic diagrams in order.

Figure 3 demonstrates a systematic process in which an underground rock mass is first subjected to high pressure, resulting in the formation of a large number of microfractures. The interface between the two sides of a fault develops into a state that is ready to slip and rupture further; thus, once the stress that is continuously imposed on the fault reaches a threshold, an EQ is triggered. Therefore, once an Ez precursor appears, it signifies that the whole fault is becoming unstable. Observational experience suggests that the duration of time between this occurrence and the onset of an EQ is only approximately several hours to one day.

The above process illustrated in Figure 3 describes a possible hourly scale critical state. The rock mass at the hypocentral depth is in a critical state during the final stage of microrupture expansion before an EQ. However, the stress field acting on the interface and fracturing the rock varies depending on different sources of stress; consequently, the time period in which an EQ occurs differs among different locations at the hourly scale. Nevertheless, all of the evidence indicates that, at this stage, the time interval over which a rock mass can remain unfractured at the hypocentral depth is approximately several hours to one day. This is likely to be a common feature of inland EQs worldwide that are triggered by the eventual cracking of rock masses under driving stresses. In addition, abundant rock fractures are interconnected at the subsurface and are prepared to slip and rupture. Furthermore, the high-temperature and high-pressure environment deep underground drives the emission of gases (such as $CO_2$, helium, radon and water vapor) [25,26] from the hypocentral depth to the atmosphere, which enhances the output of thermal radiation and causes sudden changes in the atmospheric composition and local meteorological conditions. These radioactive gases can, in turn, undergo decay and emit $\alpha$ particles [11,27–32], which may further ionize the local atmosphere within minutes. With the increasing emission of radioactive gases, many positively and negatively charged particles are injected close to the surface near the epicenter. Then, a large number of electron-ion pairs disperse in the air and cations and anions, which carry positive and negative charges, respectively, are separated by the fair-weather electrostatic static field Ez, gravity and thermal convection and are transported down and up, respectively. Then, a newly established upward electric field is formed in the atmosphere. This whole process can be completed within minutes. This illustrates that, in the final stage leading up to an EQ, there is a large release of radioactive gases such as radon near the impending fracture region. Finally, these numerous ions are separated by the fair-weather electrostatic field Ez, gravity and thermal convection, resulting in a locally polarized electric field, the direction of which opposes that of the natural electric field under fair weather; thus, these anomalous features can be easily identified. Ultimately, multipoint observations near the fracture zone can simultaneously confirm whether the signals are cosourced and facilitate their verification. Thus, the geological movement of deep rock induces an abnormal Ez signal, which is a very short-term precursor to an imminent emergency. However, there has not yet been established a large-scale monitoring network, and there are few observation sites. Moreover, the observation of meteorological conditions at the same location has not been covered. Therefore, sufficient statistical instances to support earthquake forecasting are not available.

## 5. Summary

During the final stage just before any significant EQs, drastic changes in fault movement result in the following:

(1)  There is a special critical window before an EQ to detect drastic changes in deep fault movement.
(2)  The duration of this window is 2–48 h.
(3)  During these 2–48 h, the near-surface vertical atmospheric electrostatic field always exhibits an abnormal negative signal.
(4)  At the fault near the epicenter, many anomalous signals can be observed in the surface vertical atmospheric electrostatic field.

Therefore, a negative anomalous Ez signal could be considered an important precursor of significant, imminent EQs (Ms ≥ 3). Therefore, anomalously negative Ez signals under fair weather conditions could be used as an important precursor for imminent (2–48 h prior) major inland EQs. Based on a network monitoring system and artificial intelligence technology, EQ information can be obtained very quickly. Therefore, an early (within 2–48 h) alarm for inland EQs (Ms ≥ 3) might be possible.

**Author Contributions:** T.C. designed the project and wrote the paper. L.L., X.Z. (Xiaoxin Zhang), X.J., H.W., S.W. and S.T. collected, analyzed and compared the atmospheric electrostatic field Ez data with the earthquake data. C.W. and X.Z. (Xuemin Zhang) contributed to the study in terms of space physics, atmospheric physics, atomic physics theory and signal analysis. J.S., W.L., J.L., C.C., X.P., S.C. and X.H. were responsible for building the atmospheric electrostatic field instrument, implementing the calibration work and performing the Ez measurements. All authors have read and agreed to the published version of the manuscript.

**Funding:** The authors were supported by the Strategic Pioneer Program on Space Science, Chinese Academy of Sciences (Grant Nos. XDA17010301, XDA15052500 and XDA15350201) and by the National Natural Science Foundation of China (Grant Nos. 41874176 and 41931073). The authors thank the Chinese Meridian Project, Ground-Based Space Environment Monitoring Network (Meridian Project II), Specialized Research Fund for the State Key Laboratories and the Pandeng Program of the National Space Science Center and International Partnership Program of the Chinese Academy of Sciences (Grant No. 183311KYSB20200003).

**Institutional Review Board Statement:** Not applicable.

**Data Availability Statement:** In this study, the weather data can be found on the website (https://q-weather.info) and the datasets of the atmospheric electric field are available from the corresponding author on reasonable request.

**Acknowledgments:** The authors thank Fushan Luo, Jie Liu, Zhijun Niu and Xiujie Jiang for their invaluable input and the website https://q-weather.info (accessed on 20 October 2022) for providing the meteorological data.

**Conflicts of Interest:** The authors declare no conflict of interests.

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
