# Peer review of "Possible Locking Shock Time in 2–48 Hours"

_applsci, doi:10.3390/app13020813_

Round 1

Reviewer 1 Report

1. What is the main question addressed by the research? The main issue of this article is to identify the time parameters of the earthquake precursor, based on the detection of negative anomalies of atmospheric electricity in good weather, based on regional characteristics. 2. Do you consider the topic original or relevant in the field? Does it address a specific gap in the field? The topic of searching for earthquake precursors is very relevant at the present time. Including for researchers of atmospheric electricity. Here it is necessary to take into account both meteorological factors and the seismic regime in the region. The authors showed that for their region the precursor in the electric field works 2-48 hours before the earthquake. 3. What does it add to the subject area compared with other published material? Different authors allocate the time of the precursor in different ways, based on the characteristics of the region they are exploring. The authors of this article have shown that in their region the operating time of the harbinger is 2-48 hours. This result will be taken into account by other researchers when studying the anomalies of atmospheric electricity in their regions. 4. What specific improvements should the authors consider regarding the methodology? What further controls should be considered? To determine the quality of the work of the harbinger, it is necessary to allocate the time, magnitude and location of the epicenter of the earthquake. The authors took the time. It is necessary to determine the minimum magnitude when it is possible to judge the earthquake forecast. And it is necessary to allocate the area of the future epicenter of the earthquake. Sometimes this area is a circle (Dobrovolsky's radius), sometimes a more complex figure. It depends on the geological features of the region. Strong earthquakes are quite rare. Therefore, to isolate these parameters, a long time of observations will be required. I hope the results will appear in future articles. 5. Are the conclusions consistent with the evidence and arguments presented and do they address the main question posed? The conclusions in this work are consistent with the evidence. An extensive review of the literature has been made and the conclusions are consistent with the results of other researchers. 6. Are the references appropriate? The links in this article are relevant and complete. They reflect the latest findings in this area of research. 7. Please include any additional comments on the tables and figures. The figures and table clearly present the results obtained.

Author Response

Dear editors and reviewers:

Thank you for your kind letter on our manuscript (applsci-2126949), "Possible locking shock time in 2-48 hours" on December 23, 2022. We revised the manuscript shown in red in accordance with the reviewers' comments and carefully proofread the manuscript. Here, we describe our revision according to the reviewers' comments.

Reviewer #1: 1. What is the main question addressed by the research? The main issue of this article is to identify the time parameters of the earthquake precursor, based on the detection of negative anomalies of atmospheric electricity in good weather, based on regional characteristics.

Response: [Thank you for your positive response. This article presents a possible seismic precursor signal that may be locked at 2-48 hours, which could help us avoid seismic hazards based on local geological conditions and regional characteristics in the near future, especially for strong earthquakes. The reason is that meteorological characteristics before the strong earthquake help researchers to exclude meteorological activities easier (Chen et al., 2021b).]

  1. Do you consider the topic original or relevant in the field? Does it address a specific gap in the field? The topic of searching for earthquake precursors is very relevant at the present time. Including for researchers of atmospheric electricity. Here it is necessary to take into account both meteorological factors and the seismic regime in the region. The authors showed that for their region the precursor in the electric field works 2-48 hours before the earthquake.

Response: [Thank you, as you mentioned, it is necessary to take into account both meteorological factors and other impact factors in the region before we recognize it as an earthquake precursor. For the innovation points of this paper, the anomalous atmospheric electric field signal before earthquakes has been studied by numerous researchers before, and we have observed and counted many cases of earthquake precursor signals as shown in Figure 1 and Table 1, so locking the time scale between the anomalous signal and earthquakes with 2-48 hours is the main innovation of this paper. ]

  1. What does it add to the subject area compared with other published material? Different authors allocate the time of the precursor in different ways, based on the characteristics of the region they are exploring. The authors of this article have shown that in their region the operating time of the harbinger is 2-48 hours. This result will be taken into account by other researchers when studying the anomalies of atmospheric electricity in their regions.

Response: [We appreciate your comments. As you say, different authors allocate the time of the precursor in different ways based on the regional characteristics. For example, Choudhury et al. (2013) described the characteristics of the vertical atmospheric electrostatic field as negative 7-12 hours before an EQ according to the statistics of 30 EQ events of various classes over northern India, and Smirnov (2008) proposed that the probability of earthquake prediction over 24 h before an earthquake based on the vertical atmospheric electrostatic field anomaly is 36%. Based on our statistical results and observations, we lock the negative atmospheric electric field anomaly signal at 2-48 hours before an earthquake.]

  1. What specific improvements should the authors consider regarding the methodology? What further controls should be considered? To determine the quality of the work of the harbinger, it is necessary to allocate the time, magnitude and location of the epicenter of the earthquake. The authors took the time. It is necessary to determine the minimum magnitude when it is possible to judge the earthquake forecast. And it is necessary to allocate the area of the future epicenter of the earthquake. Sometimes this area is a circle (Dobrovolsky's radius), sometimes a more complex figure. It depends on the geological features of the region. Strong earthquakes are quite rare. Therefore, to isolate these parameters, a long time of observations will be required. I hope the results will appear in future articles.

Response: [Thank you for your useful suggestions. As you mentioned, the epicenter, minimum magnitude and occurrence time of the earthquake are the three important elements. We will monitor near-ground anomalously negative atmospheric electrostatic field signals by establishing a network in the near future. The location of the epicenter could be determined by the atmospheric electrostatic field data gradient from multiple stations and calibration based on local geological conditions. The magnitude of an earthquake is derived from Dobrovolsky's radius; as you mentioned, this area might be a circle because most of them are distributed along earthquake fracture faults. We will consider these factors and make long-term observations by establishing a monitoring network.]

  1. Are the conclusions consistent with the evidence and arguments presented and do they address the main question posed? The conclusions in this work are consistent with the evidence. An extensive review of the literature has been made and the conclusions are consistent with the results of other researchers. 6. Are the references appropriate? The links in this article are relevant and complete. They reflect the latest findings in this area of research. 7. Please include any additional comments on the tables and figures. The figures and table clearly present the results obtained.

Response: [We appreciate your positive comments.]

We tried our best to improve the manuscript and made some changes in the manuscript. We earnestly appreciate the Editors/Reviewers’ work and hope that the corrections will be met with approval. Once again, thank you very much for your comments and suggestions.

Reviewer 2 Report

The text refers to the use of anomalously negative atmospheric electrostatic signal data as an earthquake precursor. To prove the relationship between this parameter and the occurrence of earthquakes, it presents the statistics of several earthquakes in China that show this earthquake precursor. The topic is novel and attractive, and the text is fluent; but before accepting for publication, it needs to address the following comments.

Comments

1.     One of the problems of predicting earthquakes is the social consequences of the wrong prediction. In this research, no statistics were reported from the abnormal atmospheric electrostatic signal cases that were recorded but no subsequent earthquake occurred (false indicator). It is necessary to provide statistics that answer this question.

2.     Which instrument does record the atmospheric electrostatic signal? Has this parameter been measured in different regions of China since the 1990s? Please, add a clause in the article to describe the measurement tools of these changes and the characteristics of the measurement stations.

3.     Please, answer in the text, how many hours after the abnormality is registered, the earthquake occurrence is possible? Because in the event of an unsuccessful prediction, it is necessary for people who have left their work and home to know how long they should stay in a safe place. This is effective in the amount of insurance for material damage and business interruption due to the notification of danger alarms.

4.     Figure 1: The text and numbers are too small. Please make the text bigger.

5.     Line 153: Delete the letter "A".

6.     Line 164: The phrase "This To" should be corrected at the beginning of the sentence.

Author Response

Dear editors and reviewers:

Thank you for your kind letter on our manuscript (applsci-2126949), "Possible locking shock time in 2-48 hours" on December 23, 2022. We revised the manuscript shown in red in accordance with the reviewers' comments and carefully proofread the manuscript. Here, we describe our revision according to the reviewers' comments.

Reviewer #2: The text refers to the use of anomalously negative atmospheric electrostatic signal data as an earthquake precursor. To prove the relationship between this parameter and the occurrence of earthquakes, it presents the statistics of several earthquakes in China that show this earthquake precursor. The topic is novel and attractive, and the text is fluent; but before accepting for publication, it needs to address the following comments.

Response: [Thank you for your positive comments, and we have improved the article according to your useful suggestions.]

Comments

  1. One of the problems of predicting earthquakes is the social consequences of the wrong prediction. In this research, no statistics were reported from the abnormal atmospheric electrostatic signal cases that were recorded,but no subsequent earthquake occurred (false indicator). It is necessary to provide statistics that answer this question.

Response: [Thank you for your useful comments. We also focus on the false alarm rate, which is very important and influential. However, we have not yet established a large-scale monitoring network, and there are few observation sites. Moreover, the observation of meteorological conditions at the same location has not been covered. Therefore, there are not enough statistical instances to be able to support earthquake forecasting, especially the false alarm rate. Most of the precursor signal statistics are conducted after earthquakes in China and other countries, but we are confident that the false alarm rate of warning earthquakes using the atmospheric electric field monitoring network is low. ]

  1. Which instrument does record the atmospheric electrostatic signal? Has this parameter been measured in different regions of China since the 1990s? Please, add a clause in the article to describe the measurement tools of these changes and the characteristics of the measurement stations.

Response: [Thank you for the suggestion, and we have added a clause in the revision to describe the measurement tools as shown in lines.]

  1. Please, answer in the text, how many hours after the abnormality is registered, the earthquake occurrence is possible? Because in the event of an unsuccessful prediction, it is necessary for people who have left their work and home to know how long they should stay in a safe place. This is effective in the amount of insurance for material damage and business interruption due to the notification of danger alarms.

Response: [We appreciate your detailed comments. First, for small and medium earthquakes, we only use them for research, and earthquake warnings are mainly aimed at strong earthquakes (Ms ≥ 6), which are rare. Based on our current statistics, most of the negative atmospheric electric field anomaly signals appeared a dozen hours before the earthquake, and most of them did not exceed 24 hours. Therefore, this is a maximum loss of 24 hours for people for most cases.]

  1. Figure 1: The text and numbers are too small. Please make the text bigger.

Response: [We apologize for the previous small text in Figure 1, and we have improved this in the revised version.]

  1. Line 153: Delete the letter "A".

Response: [Thank you; it has been corrected.]

  1. Line 164: The phrase "This To" should be corrected at the beginning of the sentence.

Response: [Thank you for your detailed comments; we have corrected it. ]

We tried our best to improve the manuscript and made some changes in the manuscript. We earnestly appreciate the Editors/Reviewers’ work and hope that the corrections will be met with approval. Once again, thank you very much for your comments and suggestions.

Round 2

Reviewer 2 Report

Most of the comments are correctly addressed in the text. The only remaining comment is comment 1. Please add your response in a suitable place in the text: "It is also important to focus on the false alarm rate. However, it has not yet established a large-scale monitoring network, and there are few observation sites. Moreover, the observation of meteorological conditions at the same location has not been covered. Therefore, sufficient statistical instances to support earthquake forecasting are not available, especially the false alarm rate. Most of the precursor signal statistics are conducted after earthquakes. Meanwhile, it is confident that the false alarm rate of warning earthquakes using the atmospheric electric field monitoring network is low."

Author Response

Dear editors and reviewers:

Thank you for your kind letter on our manuscript (applsci-2126949), "Possible locking shock time in 2-48 hours" on December 28, 2022. We revised the manuscript shown in red in accordance with the reviewers' comments and carefully proofread the manuscript. Here, we describe our revision according to the reviewers' comments.

Reviewer #2: Most of the comments are correctly addressed in the text. The only remaining comment is comment 1. Please add your response in a suitable place in the text: "It is also important to focus on the false alarm rate. However, it has not yet established a large-scale monitoring network, and there are few observation sites. Moreover, the observation of meteorological conditions at the same location has not been covered. Therefore, sufficient statistical instances to support earthquake forecasting are not available, especially the false alarm rate. Most of the precursor signal statistics are conducted after earthquakes. Meanwhile, it is confident that the false alarm rate of warning earthquakes using the atmospheric electric field monitoring network is low."

Response: [Thank you for your detailed comments, and we have improved the article as shown in lines 262-265 according to your useful suggestions.]

We tried our best to improve the manuscript and made some changes in the manuscript. We earnestly appreciate the Editors/Reviewers’ work and hope that the corrections will be met with approval. Once again, thank you very much for your comments and suggestions.
